# Rotation-Equivariant Conditional Spherical Neural Fields for Learning a Natural Illumination Prior

**James A. D. Gardner**
Department of Computer Science
University of York
York, United Kingdom
james.gardner@york.ac.uk

**Bernhard Egger**
Cognitive Computer Vision Lab
Friedrich-Alexander-Universität
Erlangen, Germany
bernhard.egger@fau.de

**William A. P. Smith**
Department of Computer Science
University of York
York, United Kingdom
william.smith@york.ac.uk

## Abstract

Inverse rendering is an ill-posed problem. Previous work has sought to resolve this by focussing on priors for object or scene shape or appearance. In this work, we instead focus on a prior for natural illuminations. Current methods rely on spherical harmonic lighting or other generic representations and, at best, a simplistic prior on the parameters. We propose a conditional neural field representation based on a variational auto-decoder with a SIREN network and, extending Vector Neurons, build equivariance directly into the network. Using this, we develop a rotation-equivariant, high dynamic range (HDR) neural illumination model that is compact and able to express complex, high-frequency features of natural environment maps. Training our model on a curated dataset of 1.6K HDR environment maps of natural scenes, we compare it against traditional representations, demonstrate its applicability for an inverse rendering task and show environment map completion from partial observations. A PyTorch implementation, our dataset and trained models can be found at jadgardner.github.io/RENI.

## 1   Introduction

Compact but expressive lighting representations play an essential role in graphics, enabling realistic lighting effects at real-time frame rates [38, 53, 54, 34, 22] and in computer vision enabling scene relighting [59, 58], face relighting [55, 43, 17, 42] and object insertion [56, 47, 31]. Real-world illumination is highly complex and variable, with a very high dynamic range, and is therefore inherently challenging to represent. However, real-world illumination does contain statistical regularities [16], particularly for outdoor, naturally lit scenes. For example, lighting usually comes predominantly from above, often with the strongest illumination coming from primary light sources in a few directions. Also, the sky and sunlight produce only a limited range of colours. In addition, illumination environments have a canonical up direction (vertical axis aligns with gravity) but arbitrary horizontal rotation (any rotation about the vertical is equally likely). These regularities and geometric symmetries can significantly restrict the space of possible illuminations to constrain inverse problems or enable the synthesis of realistic lighting.

36th Conference on Neural Information Processing Systems (NeurIPS 2022).

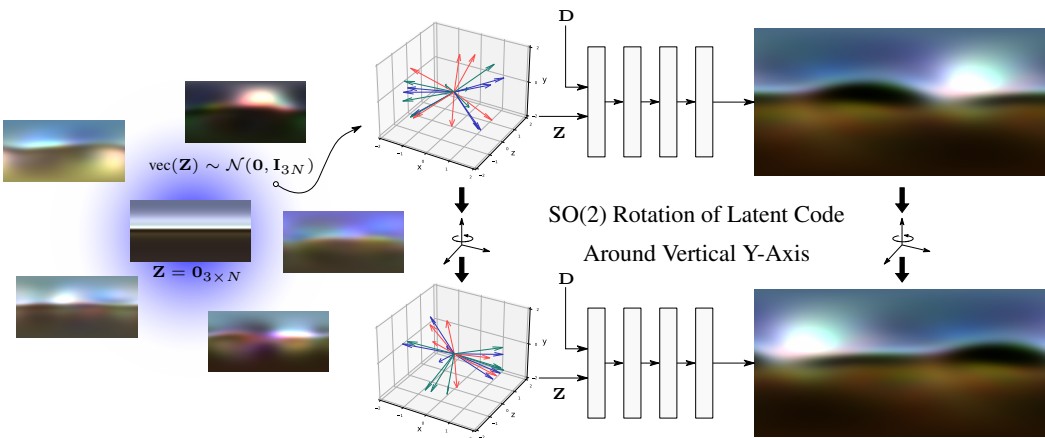

Figure 1: On the left, we visualize environment maps derived from random latent samples of RENI, our natural illumination prior, as well as the average illumination in the centre. RENI is rotation-equivariant to rotations of the latent codes around the vertical $y$-axis (right). Plots are shown for a $3 \times 20$ latent code at two rotations, 160 degrees apart, and the resulting output of the RENI network equally rotated.

**Lighting Representations**  An illumination environment is a spherical signal. A relatively small set of alternatives are used for their representation within vision and graphics. A widely used representation in graphics is an environment map [39, 38, 53, 54], which is a regularly sampled 2D image representing a flattening of the sphere, usually via an equirectangular projection. However, the projection introduces distortions leading to irregular sampling on the sphere, it is not compact, introduces boundaries and provides no constraint. Nevertheless, environment map representations have been used in inverse settings where every pixel in the map is optimised independently [41].

Spherical harmonic (SH) lighting [3, 38] is a compact lighting representation commonly used in real-time computer graphics [38, 22, 46] and inverse rendering [59, 53, 31, 40, 17]. While SHs can be used to represent the illumination environment directly, more commonly, they represent pre-integrated lighting, i.e. the illumination environment convolved with a bidirectional reflectance distribution function (BRDF). When the BRDF is low frequency, as it is for Lambertian diffuse reflectance, then the convolution is also low frequency making the approximation with SHs very accurate [3].

An alternative, growing in popularity, is the Spherical Gaussian (SG) representation [54, 53, 61]. SGs represent a lighting environment as a collection of Gaussian lobes on the sphere, each of which has 6 degrees of freedom (three for RGB amplitude, two for spherical direction and one for sharpness). While this allows the reconstruction of localised high-frequency features, it still requires many lobes to approximate complex illumination environments. [31] compares SH and SG for object re-lighting, finding SG was able to recover higher frequency lighting using a similar number of parameters as SH, though both still required a large number of parameters to approximate ground truth.

Both SHs and SGs are *rotation equivariant*. A rotation of the illumination environment corresponds directly to a rotation of the SH basis or the SG lobe directions. Equivalently, they can represent any rotation of a given environment with equal accuracy. However, they provide no prior over the space of possible illuminations. SGs or SHs can represent any colour of light coming from any direction.

**Illumination Priors**  When humans solve inverse rendering tasks, they rely on strong priors over the space of possible illuminations. For example, they resolve convex/concave shading ambiguities with the lighting-from-above assumption [26, 52]. Given this, it is surprising that statistical illumination priors have been almost completely ignored in computer vision, with the vast majority of inverse rendering techniques allowing arbitrary illumination within their chosen representation space.

There are a small number of exceptions. Both Egger et al. [17] and Yu and Smith [59] learn a linear statistical model with Gaussian prior in the space of SH coefficients. While providing a useful constraint to avoid unrealistic illumination environments, this approach inherits the weakness of SHs in being unable to reproduce high-frequency lighting effects while also losing the rotation

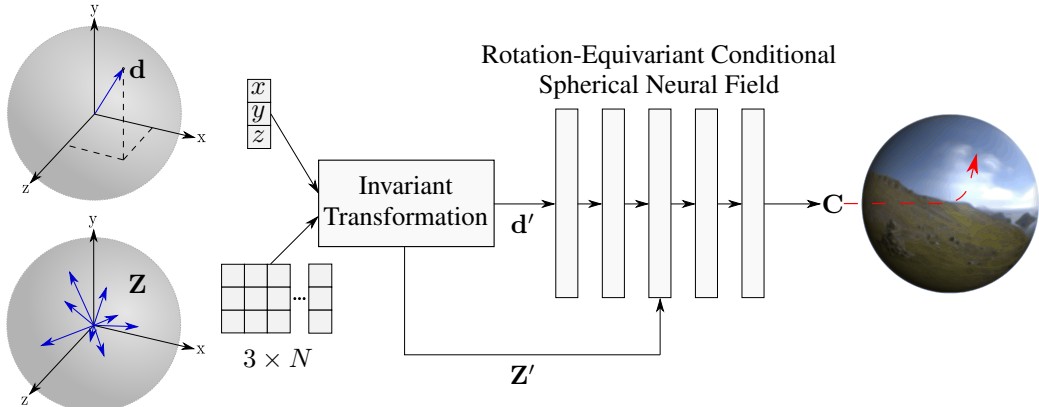

Figure 2: We propose to represent a space of spherical signals via a rotation-equivariant conditional spherical neural field. The signal in a direction **d** can be queried by evaluation of the network and rotating the Vector Neuron conditioning latent code **Z**, corresponds to rotating the spherical signal.

equivariance. Yu and Smith [59] seek to overcome this by rotation augmentation at training time, but this brute force approach makes no guarantee of rotation equivariance. Sztrajman et al. [49] separate environment maps into HDR and LDR components. Using a CNN-based auto-encoder for estimations of LDR components of lighting alongside a low dimensional SG model for the HDR lighting provided by the sun. They too require data augmentation in the form of rotations, can make no equivariance guarantees and, due to the low dimensionality of the SG model, struggle to represent environments with multiple HDR light sources.

**Neural Fields**   Neural fields [57] have provided impressive results in a range of applications including representations of objects and scenes [44, 2, 33, 13, 12, 23, 36, 50], in inverse rendering [7, 5, 40, 48, 8, 61] and robotics [30, 11, 35]. The work most closely related to ours is Neural-PIL [8]. They use a FiLM SIREN similar to that proposed in Pi-GAN [10], and like us use a direction query vector and auto-decoder architecture. However, they do not have rotation-equivariance and apply no natural light prior. They also model pre-integrated lighting, conditioning the latter layers of their FiLM SIREN on a material roughness parameter.

**Rotation Invariance/Equivariance**   Two important symmetries in computer vision are invariance and equivariance to the rotation group [9]. Some works attempt to achieve these properties via data augmentation [59, 37, 49], which still results in missed cases for continuous rotations. A solution alleviating the need for extensive data augmentation is the Vector Neuron [15], which offers a framework for designing SO(3)-Equivariant networks via a latent matrix representation rather than a latent vector. This allows a direct mapping of rotations applied to the network's input to its output, resulting in all possible rotations being explicitly represented via rotations of the latent codes.

**Contribution**   We desire a representation of natural illumination environments that offers the following features. *Generative:* A generative model that captures the statistical regularities of natural illumination with a well-behaved latent space within which we can optimise to solve inverse problems. *Compact:* Reduces the dimensionality of inverse problems while preserving high-frequency lighting effects that are important for non-Lambertian appearance. *Rotation-Equivariant:* Respect the canonical orientation, i.e. any rotation of an environment about the vertical should be equally likely and equally well represented. *Statistical Prior:* Provides a prior to regularise inverse problems, or that can be sampled from for synthesis, only generating plausible illumination environments. *HDR:* Correctly handle HDR quantities essential for realistic rendering and the reproduction of natural light.

We introduce RENI - A Rotation-Equivariant Natural Illumination model. Our key contributions are:

- An extension of Vector Neurons to a rotation-equivariant neural field representation for spherical images, optionally restricted to rotations about the vertical axis.
- A variational autodecoder architecture for a generative model of spherical signals.

– The first natural, outdoor HDR illumination model.

– Evaluated in an inverse rendering task showing significant performance improvements over other lighting representations.

We choose to model scene radiance, i.e. environment lighting, directly as opposed to pre-integrated lighting with a particular BRDF. This makes our model more general since it can be used with arbitrary BRDFs at inference time or even for tasks other than rendering, such as to constrain shape from specular flow.

## 2  Rotation-Equivariant Conditional Spherical Neural Fields

We wish to construct a generative model of spherical signals that is rotation equivariant with respect to the latent representation of the signal. That is to say, a rotation of the latent representation corresponds to a rotation of a spherical signal and a signal can be reconstructed with exactly the same accuracy in any rotation. We begin by proposing a variant of Vector Neurons [15] for $SO(3)$ equivariant representation of spherical signals.

To achieve this, we use an ordered list of 3D vectors for our latent representation. Our model takes the form of a conditional spherical neural field, $f : S^2 \times \mathbb{R}^{3 \times N} \to \mathbb{R}^M$, such that $f(\mathbf{d}, \mathbf{Z})$ computes the value of the signal represented by latent code $\mathbf{Z} \in \mathbb{R}^{3 \times N}$ in direction $\mathbf{d} \in \mathbb{R}^3$, with $\|\mathbf{d}\| = 1$. For colour images, $M = 3$ and $f$ outputs an RGB colour. By using a spherical neural field, we are agnostic to how the signals are sampled on the sphere. We can generate any sampling simply by choosing the grid of directions as appropriate. We also avoid boundary effects since our domain is continuous.

We construct the neural field such that it is *invariant* to a rotation of both $\mathbf{d}$ and $\mathbf{Z}$ simultaneously (i.e. $f(\mathbf{Rd}, \mathbf{RZ}) = f(\mathbf{d}, \mathbf{Z})$ with $\mathbf{R} \in SO(3)$). This entails that the neural field is *equivariant* with respect to a rotation of $\mathbf{Z}$ only (i.e. rotating $\mathbf{Z}$ corresponds to rotating the spherical signal such that $f(\mathbf{d}, \mathbf{RZ}) = f(\mathbf{R}^\top \mathbf{d}, \mathbf{Z})$).

**Equivariant Transformation**   Key to our approach is a transformation of the inputs to the neural field, $(\mathbf{d}, \mathbf{Z})$, such that they are rotation invariant. These divide into two parts: 1. $\mathbf{d}'$ - the directional input to the spherical neural field and 2. $\mathbf{Z}'$ - the latent code on which the neural field is conditioned.

The direction in which we wish to evaluate the spherical neural field must be encoded relative to the latent code in the particular rotation in which we encounter it. This is satisfied by using the inner product $\langle \mathbf{d}, \mathbf{Z} \rangle$, i.e. the matrix-vector product $\mathbf{d}' = \mathbf{Z}^\top \mathbf{d} \in \mathbb{R}^N$. Unlike Vector Neurons, our input is a direction not a position. Hence, the rotation invariant feature $\|\mathbf{d}\|$ conveys no information and we do not use it.

For the latent code, we follow the Vector Neurons invariant layer: $\mathbf{Z}' = \text{VN-In}(\mathbf{Z})$. However, we use the full Gram matrix $\mathbf{Z}^\top \mathbf{Z}$ since we expect the dimensionality of the latent space, $N$, to remain moderate for our data. If this $O(N^2)$ size becomes problematic, we can use the same scalable solution used by Vector Neurons [15].

Since our neural field is equivariant **we do not need to augment our training data over the space of rotations**. Observing a spherical signal once means we can reconstruct it with the same accuracy in any rotation.

**Variational Auto-decoder**   We train our conditional spherical neural field as a decoder-only architecture, i.e. an auto-decoder [36]. This means that we optimise the network weights simultaneously with the latent codes for each training sample. This avoids the need to design a rotation-equivariant encoder while, for inverse tasks, only the decoder is needed so we avoid the redundancy of also training an encoder. However, training with no regularisation on the learnt latent space does not lead to a space that is smooth or that follows a known distribution. This means it cannot be sampled from, does not produce meaningful interpolations and provides no prior for inverse problems. For this reason, we use a *variational auto-decoder* [60] architecture. Each training sample is represented by a mean, $\boldsymbol{\mu}_i \in \mathbb{R}^{3N}$, and standard deviation, $\boldsymbol{\sigma}_i \in \mathbb{R}^{3N}$, that provide the parameters of a normal distribution from which the flattened latent code for that training sample is drawn: $\text{vec}(\mathbf{Z}_i) \sim \mathcal{N}(\boldsymbol{\mu}_i, \boldsymbol{\Sigma}_i)$, where $\boldsymbol{\Sigma}_i = \text{diag}(\sigma_{i,1}^2, \ldots, \sigma_{i,3N}^2)$ is the diagonal covariance matrix. Using the reparameterisation

trick [28], we can generate a latent code $\text{vec}(\mathbf{Z}_i) = \boldsymbol{\mu}_i + \boldsymbol{\sigma}_i \odot \boldsymbol{\epsilon}$ where the noise is sampled as $\boldsymbol{\epsilon} \sim \mathcal{N}(\mathbf{0}, \mathbf{I}_{3N})$. During training, we optimise $\boldsymbol{\mu}_i$ and $\boldsymbol{\sigma}_i$ for each training sample and use the Kullback–Leibler divergence as a loss to regularise the distribution of each latent code toward the standard normal distribution:

$$\mathcal{L}_{\text{KLD}} = -\frac{1}{2} \sum_{i=1}^{K} \sum_{j=1}^{3N} (1 + \log(\sigma_{i,j}^2) - \mu_{i,j}^2 - \sigma_{i,j}^2), \tag{1}$$

where $K$ is the number of training samples.

## 3   RENI: A Statistical Model of Natural Illumination

We now describe how to construct a statistical model of natural illumination environments as a restricted version of a rotation-equivariant conditional spherical neural field. We call our model RENI (Rotation-Equivariant Natural Illumination). In contrast to the full $SO(3)$-equivariant formulation in Section 2, we propose a variant with only $SO(2)$ equivariance and explain in detail our training data, losses and implementation.

**SO(2) Equivariance**   Natural environments have a canonical "up" direction (defined by gravity) but arbitrary rotation about this vertical axis. For this reason, we do not want arbitrary $SO(3)$ rotation equivariance. This would have the undesirable effect of permitting unnatural environment orientations (such as with the sky at the bottom), providing a less useful prior when solving inverse problems. So, we propose a restricted transformation of the neural field inputs that are invariant only to rotations, $\mathbf{R}_y$, about the vertical ($y$) axis.

In order to construct the invariant features, we use two selection matrices: $\mathbf{S}_{xz}$ selects the components that are orthogonal to the axis of rotation (i.e. the $x$ and $z$ components) and are thus affected by a $y$-axis rotation, $\mathbf{s}_y$ selects the unaffected $y$ component:

$$\mathbf{S}_{xz} = \begin{bmatrix} 1 & 0 & 0 \\ 0 & 0 & 1 \end{bmatrix}, \quad \mathbf{s}_y = \begin{bmatrix} 0 & 1 & 0 \end{bmatrix}$$

The directional part of the invariant input now contains three components: $\mathbf{d}' = (\mathbf{s}_y \mathbf{d}, \langle \mathbf{S}_{xz} \mathbf{d}, \mathbf{S}_{xz} \mathbf{Z} \rangle, \|\mathbf{S}_{xz} \mathbf{d}\|)$. The first is the invariant component of $\mathbf{d}$. The second encodes $\mathbf{d}$ relative to $\mathbf{Z}$ in the $x$-$z$ plane. The third measures the norm of $\mathbf{d}$ projected into the $x$-$z$ plane which is unchanged by rotations about $y$.

The conditioning part of the invariant input now contains two components: $\text{VN-In}(\mathbf{Z}) = \mathbf{Z}' = (\mathbf{s}_y \mathbf{Z}, (\mathbf{S}_{xz} \mathbf{Z})^\top \mathbf{S}_{xz} \mathbf{Z})$. The first is simply the invariant component of each column of $\mathbf{Z}$. The second is the invariant transformation (Gram matrix) of latent vectors projected into the $x$-$z$ plane.

**Reconstruction Loss and HDR**   HDR imaging is essential for realistic rendering and enables the accurate representation of the full dynamic range of natural light. Therefore our model must learn an HDR representation of natural illumination. Computing a reconstruction loss in linear HDR space is dominated by large values and leads to poor reconstruction of most of the environment. Therefore, similar to [18], we train our network to output $\log(\text{HDR})$ values and compute the reconstruction loss in log space. We further normalise the $\log(\text{HDR})$ values to the range $[-1, 1]$ by scaling and shifting using the maximum and minimum values across the whole training set.

Each training sample comprises $P$ pairs of directions and corresponding normalised $\log(\text{HDR})$ RGB colours that we store in the matrices $\mathbf{D}_i = [\mathbf{d}_{i1}, \ldots, \mathbf{d}_{iP}] \in \mathbb{R}^{3 \times P}$ and $\mathbf{C}_i = [\mathbf{c}_{i1}, \ldots, \mathbf{c}_{iP}] \in \mathbb{R}^{3 \times P}$ respectively. RENI is agnostic to the resolution and sampling of the spherical signal. The neural field can be queried for any direction. In practice, our dataset contains spherical images in an equirectangular sampling containing $P = 2H^2$ pixels where $H$ is the height of the images. Since equirectangular images are irregularly sampled, we weight the mean squared reconstruction loss by the sine of the polar angle, $\theta(\mathbf{d})$:

$$\mathcal{L}_{\text{recon}} = \sum_{i=1}^{K} \frac{1}{P} \sum_{j=1}^{P} \sin(\theta(\mathbf{d}_{ij})) \|f(\mathbf{d}_{ij}) - \mathbf{c}_{ij}\|^2 \tag{2}$$

**Neural Field**   Our neural field, $f$, is implemented as a SIREN [44] - an MLP with a periodic activation function. This architecture has proven highly effective at representing a wide range of complex natural signals. In order to condition the SIREN on the invariant latent code $\mathbf{Z}'$, we use *conditioning-by-concatenation* [44, 57], i.e. we simply input both $\mathbf{d}'$ and $\mathbf{Z}'$ to the network. We also tested a FiLM-conditioned SIREN [10] but could not get significant performance improvements and, perhaps due to the small size of our dataset, FiLM conditioning resulted in a non-smooth latent space that could drastically affect performance when fitting to unseen images.

**Training Data**   We have curated a dataset of 1,694 HDR equirectangular images of outdoor, natural illumination environment obtained with either CC0 1.0 Universal Public Domain Dedication license [24, 19, 27] or with written permission to redistribute a low-resolution version of their dataset [25, 1, 45, 51]. All images were then checked to ensure they did not contain any personally identifiable information or offensive content and any images that contained predominantly unnatural light sources were removed. 21 images were also selected and held back for optimising only the latent codes at test time, resulting in a training dataset of 1,673 HDR images.

**Training**   We use Adam [29] to optimise the sum of the reconstruction and KLD losses:

$$\mathcal{L}_{\text{Train}} = \mathcal{L}_{\text{recon}} + \frac{\beta}{D} \mathcal{L}_{\text{KLD}} \tag{3}$$

where $\beta$ is a hyperparameter to weight the KLD loss and $D = 3N$ is the dimensionality of the latent space, used to normalise for the choice of $N$. We randomly initialise the mean latent code for each image, $\boldsymbol{\mu}_i$, from a standard normal distribution. In order to ensure the positivity of the variances, $\sigma_{i,j}^2$, we optimise $\log(\sigma_{i,j}^2)$ which we initialise randomly from a normal distribution with mean -5 and variance 1. We use all the pixels from a single training image as a mini-batch and use batch size 1.

In order to speed up training, we employ a progressive strategy. We start by training with low resolution ($H = 16$) images and double resolution every 800 epochs until we reach $H = 128$ resolution for a total of 2,400 epochs. This allows the network to quickly learn low-frequency features, gradually obtaining higher frequency details later in the training. Unlike with a convolutional architecture, the spatially-continuous nature of the neural field means we can implement multi-resolution training using a single network. We simply upsample the grid of directions, $\mathbf{D}_i$, and increase the resolution of the target equirectangular images to match this resolution.

We used Weights and Biases [6] for experiment tracking, visualisations and hyper-parameter grid search. This yielded the best performance using a variational auto-decoder, SIREN with 5 layers each with 128 hidden features. We use an exponentially decaying learning rate starting at $10^{-5}$ and decreasing to $10^{-7}$ by the end of training and weight the KLD loss $\beta = 10^{-4}$ to be of similar magnitude to the reconstruction loss. Training RENI with a latent code dimension of $D = 27$, took around 15 hours on a local Nvidia A40 48GB GPU.

**Model Fitting**   At test time, the SIREN network is held static, and only latent codes are optimised in order to fit to an unseen image. We initialise the latent code to zeros, corresponding to the mean environment map (see Figure 1, left). This provides an unbiased initialisation in the absence of any prior information about the environment. We found performance on test images was improved by including a cosine similarity loss $\mathcal{L}_{Cosine}$ on the RGB colour vectors [32] weighted using the same $\sin(\theta(\mathbf{d}))$ weight as used in $\mathcal{L}_{\text{recon}}$. We also include a prior loss on the latent vector $\mathcal{L}_{\text{prior}} = \|\mathbf{Z}\|_{\text{Fro}}^2$. Our test time loss is therefore:

$$\mathcal{L}_{\text{Test}} = \mathcal{L}_{\text{recon}} + \rho \mathcal{L}_{\text{Cosine}} + \gamma \mathcal{L}_{\text{Prior}}. \tag{4}$$

We again use a multi-resolution training scheme, fitting with the same resolutions and number of epochs as used during training. A hyper-parameter grid search found the best performance when using $\rho = 10^{-4}$, $\gamma = 10^{-7}$, and an exponentially decaying learning rate starting at $10^{-2}$ and decreasing to $10^{-4}$.

A PyTorch implementation, our dataset and trained models can be found at jadgardner.github.io/RENI.

# 4 Evaluation

**Generalisation** We begin by evaluating the generalisation performance of RENI when approximating unseen environments. We compare against SH and SG and explore how generalisation performance varies as a function of latent code dimension. We consider RGB spherical harmonics of order 2, 5, 6 and 9 equal to latent code dimension $D = 3 \times N$ for $N = 9, 36, 49, 100$. Since SG requires a dimensionality that is a multiple of 6, where $D$ is not a multiple of 6, we bias in SGs favour by using the next multiple, i.e. $\lceil D/6 \rceil \cdot 6$. We used an open-source implementation of per-pixel environment map fitting provided by [31] to fit our SG models. As shown in Figure 3, RENI can capture higher frequency detail than both SH and SG, is less dominated by high-value pixels and can reproduce accurate HDR values. A comparison of the mean PSNR across the test set images for increasing latent code dimensionality is shown in Table 1.

Table 1: The mean PSNR when fitting to the test set for increasing latent dimensions. Where an exact dimensionality comparison is not possible for SG the dimensionality used is shown in brackets.

| $D$ | RENI | SH | SG | |
|---|---|---|---|---|
| 27 | **17.02** | 12.80 | 16.04 | $(N = 30)$ |
| 108 | **19.58** | 15.89 | 18.44 | |
| 147 | **19.97** | 16.44 | 19.26 | $(N = 150)$ |
| 300 | **20.47** | 17.07 | 20.02 | |

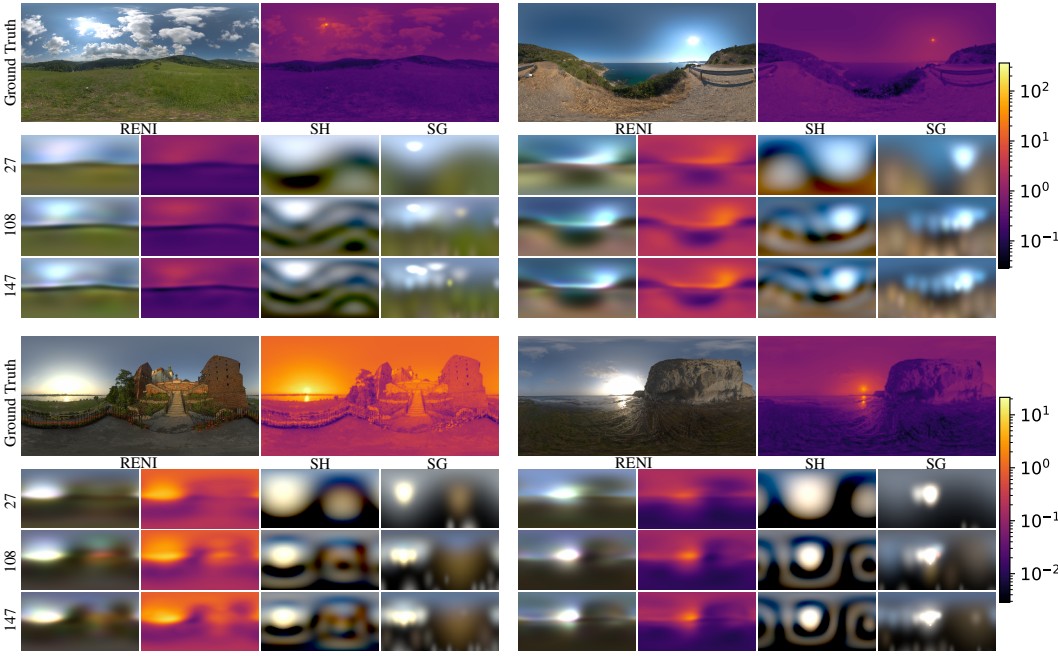

Figure 3: Generalisation to unseen images with latent code dimensions, $D = 3N$ for $N = 9, 36, 49$ and for SH of equal dimensionality (orders 2, 5, and 6). SG results are with dimensionality $D = 30, 108, 150$. Heat maps with log-scale colour bars for ground truth and RENI are also shown.

To test the impact of restricting RENI's equivariance to $SO(2)$ we ran an ablation of models with $SO(3)$, $SO(2)$ and without equivariance at three sizes of latent code dimension $D$. For the model without equivariance, we augmented the dataset with rotations of the images at increments of $0.785rad$ for a training dataset size of $13384$ images. The $SO(2)$ case performs best for all latent code sizes, and both the $SO(2)$ and $SO(3)$ outperform the model trained purely using augmentation whilst using significantly less data. Results are shown in Table 2.

Table 3 shows an ablation of model sizes. When using a smaller network, reconstruction quality suffers due to the representational power of the network being reduced. Whereas the larger networks over-fit on the training data and optimising latent codes to fit unseen images becomes more challenging, perhaps due to the small size of the dataset.

Table 2: Mean PSNR on the test set for models with varying levels of equivariance.

| D | None | SO(2) | SO(3) |
|---|------|-------|-------|
| 27 | 11.32 | **17.02** | 14.00 |
| 108 | 15.85 | **19.58** | 18.27 |
| 147 | 14.64 | **19.97** | 17.45 |

Table 3: Mean PSNR on the test set for different network sizes and latent dimensionality.

| Layers | $D = 27$ | $D = 108$ | $D = 147$ |
|--------|----------|-----------|-----------|
| 3 | 16.25 | 18.29 | 18.57 |
| 5 | **17.02** | **19.58** | **19.97** |
| 7 | 16.38 | 18.13 | 18.15 |

**Latent Space Interpolation**    As shown in Figure 4, linear interpolations between codes result in smooth transitions between images and plausible natural environments for all intermediate latent codes showing how RENI encodes a meaningful internal representation of natural illumination.

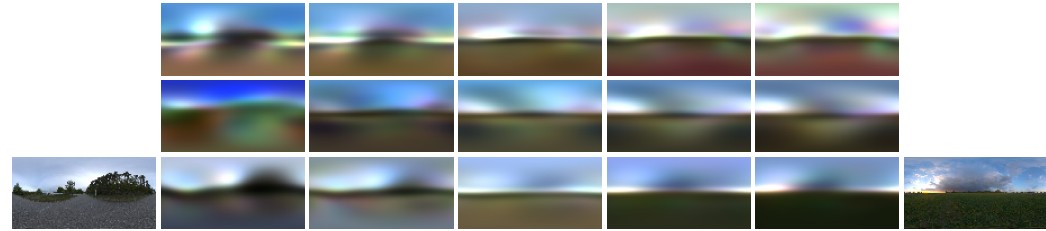

Figure 4: Interpolation results for RENI with latent code dimension of $D = 108$. Rows 1 and 2 show interpolations between two random latent codes, and row 3 shows an interpolation between two training images with the ground-truth images shown.

**Environment Completion**    Any picture of a natural scene contains cues about the surrounding environment that the scene was captured in, such as likely sun locations and possible environmental content. If RENI is only provided with a small portion of the complete environment map in its loss at test time, RENI can hallucinate plausible completions of the environment. As shown in Figure 5, RENI makes sensible estimations about the possible colours and shapes of land and sky and often predicts quite accurate sun locations despite the sun being outside the image crop.

**Inverse Rendering**    To test the performance of RENI in an inverse rendering pipeline, we implemented a normalised Blinn-Phong environment map shader in PyTorch3D, enabling fully differentiable rendering. We render a 3D object with fixed geometry, pose, camera and material parameters such that only lighting in the scene is unknown. Optimising only latent codes, we minimise the same losses as used in $\mathcal{L}_{Test}$, without the $\sin(\theta(\mathbf{d}))$ weighting needed for equirectangular images, between a rendering using a ground truth environment map and a one using the output of RENI. This was tested for incremental increases in weighting of the Blinn-Phong specular term $(K_s)$, from $K_s = 0$ to $K_s = 1.0$ in steps of 0.2. A normalisation factor $\alpha = (n + 2)/(4\pi(2 - e(\frac{-n}{2})))$ [21], was applied to $K_s$ to get a steady transition from diffuse to specular. We achieved the best performance using $\alpha = 10^3$, $\gamma = 10^{-4}$, and an exponentially decaying learning rate starting at $10^{-2}$ and decreasing to $10^{-4}$ over 2,400 epochs. We use an environment map resolution of $H = 64$ and render the object with a resolution of $128^2$. In Figure 6 we compare against SH environment maps computed in closed form using linear least squares for increasing specularity. As expected, SH performs well at producing accurate renders of diffuse objects. However, the environment maps produced are unnatural, and as the specular term increases in weight, SH degrades in performance significantly compared to RENI.

**Non-convexity of Reconstruction Error in Latent Space**    RENI is rotation equivariant. However, this does not necessarily mean that optimising to fit a rotated image will yield a rotated version of the latent code resulting from fitting to an unrotated version of the image. This means that the loss landscape of our reconstruction losses is not convex. To verify this, we fit RENI to a version of the test set containing pairs of unseen images rotated 180 degrees apart. Initialising at the mean environment and optimising resulted in pairs of latent codes $\mathbf{Z_1}, \mathbf{Z_2}$. Ideally, this would result in each latent code being explained via a $y$-axis rotation of the other. To test this we minimised $\mathbf{M}$ for $\|\mathbf{Z}_1\mathbf{M} - \mathbf{Z}_2\|_2$ and obtained a rotation matrix $\mathbf{R}$ that minimises $\|\mathbf{M} - \mathbf{R}\|_F$. We then calculate the

| Ground Truth | RENI Full | Masked Ground Truth | RENI Inpainting |
|:---:|:---:|:---:|:---:|

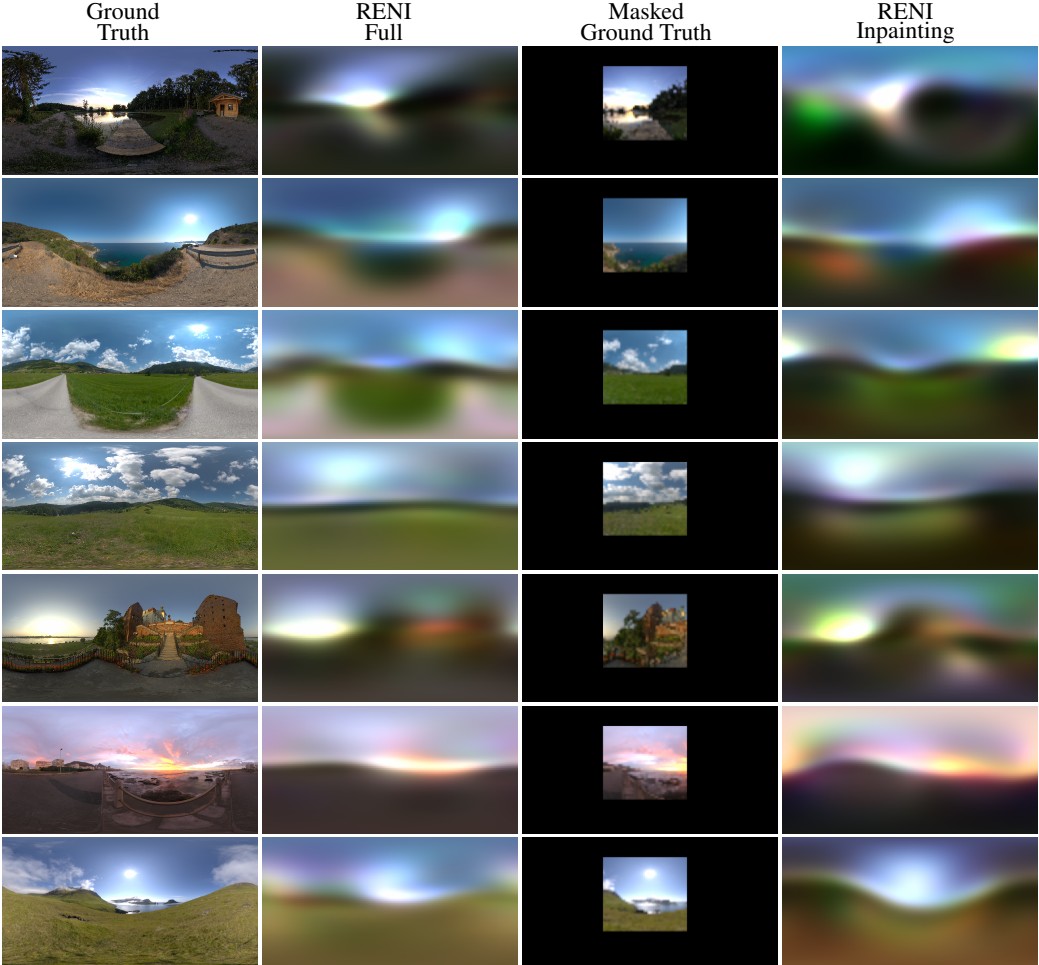

Figure 5: Col. 2 shows the output of optimising latent codes on full ground-truth (Col. 1). When trained on a masked ground-truth (Col. 3) RENI predicts plausible continuations for the environment and makes accurate estimations of sun locations (Col. 4). Results from a $D = 108$ model.

relative error between $\mathbf{R}\mathbf{Z}_1$ and $\mathbf{Z}_2$:

$$E = \frac{\|\mathbf{R}\mathbf{Z}_1 - \mathbf{Z}_2\|_F}{\|\mathbf{Z}_2\|_F}$$

For a model of $D = 27$, this resulted in $E < 2.0\%$ for 18 of the 21 image pairs, demonstrating that both latent codes can largely be explained as a simple rotation of the other. However, for the remaining three images, the error was higher, suggesting there is redundancy in the latent space, i.e. there are multiple possible explanations for a single image. Better latent space regularisation, tuning model dimensionality and a larger dataset might help resolve this.

## 5   Discussion and Conclusion

We introduced rotation-equivariant spherical neural fields and used them to create RENI, a natural illumination prior. Demonstrating how random samples from RENI always produce plausible illumination maps and RENI's usefulness for environment completion and inverse rendering. There are many exciting avenues for future research, for example, implementing RENI in larger inverse rendering pipelines where it could be a simple drop-in replacement for SH and using RENI for LDR to HDR image reconstruction. Furthermore, our spherical neural field is the initial step towards a Generative Adversarial Network [20] (GAN) for spherical signals and could be used as the generator alongside a Spherical CNN [14] for the discriminator in a GAN framework.

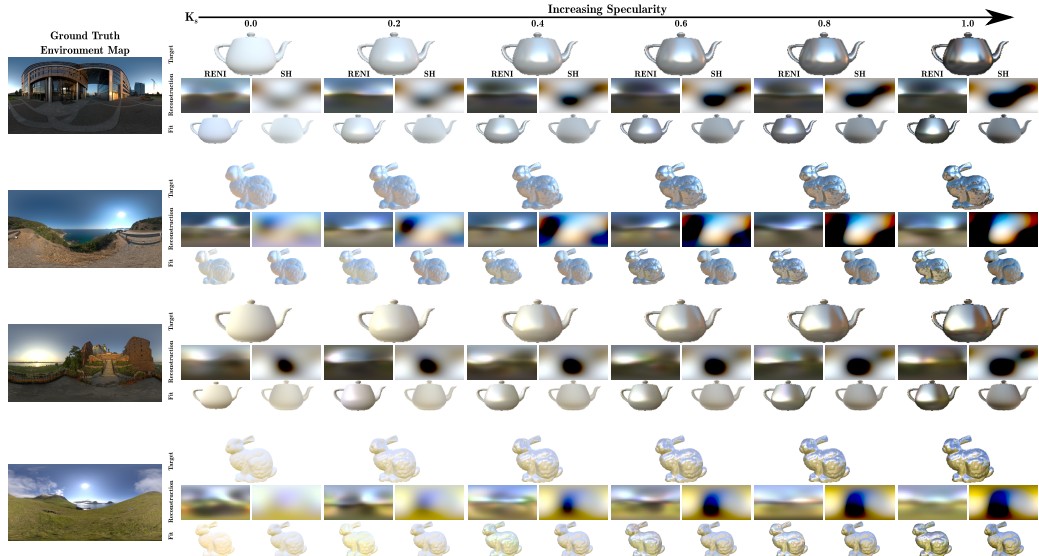

Figure 6: Reconstruction results in an inverse rendering task. The specular Blinn-Phong term $K_s$ increases from left to right in steps of $0.2$. Both RENI and SH have a dimensionality of $D = 27$. SH performs well in the purely diffuse case, but the resulting environment map reconstruction is unnatural, and as $K_s$ increases RENI performs significantly better.

**Limitations**   Using the Gram matrix for the invariant representation, which is $O(n^2)$, limits the size of the latent code you can realistically use. This could be addressed through the use of the invariant layer proposed by [15] to reduce this to $O(n)$. The very highest frequency detail is still not resolved even when using larger latent code sizes; scaling up with a larger dataset and network size is likely to address this. Because RENI has a prior for natural illumination, unlike SH, RENI's performance decreases when fit to indoor scenes. Interestingly, RENI still generalises to indoor scenes quite well; see supplementary material for examples of RENI fitting to indoor scenes.

Human vision has complex interactions between illumination, geometry and texture priors. For example, the Hollow Face Illusion [26] arises from face geometry priors overriding the lighting from above illumination prior; while in the Bas relief ambiguity [4], geometric priors cause incorrect lighting estimation. Our model discounts these interactions, learning an illumination prior independently but not its interactions with other cues.

We strictly allow only $SO(2)$ equivariance. However, considering typical camera coordinate systems, the up axis will sometimes not align with gravity when the camera is pointed up or down. For inverse problems, this would mean the gravity vector would need to be explicitly estimated (an accelerometer would resolve this). Alternatively, we could build our model with full $SO(3)$ equivariance but then learn a prior over the space of camera poses relative to gravity.

**Broader Impact**   The illumination prior presented here has the potential to improve the performance of inverse rendering pipelines, of which there are both positive and negative downstream use cases that should be considered; for example, the generation of highly realistic 3D models of a person's likeness without their consent. Advances in inverse and neural rendering could potentially lead to employment loss in the generation of 3D assets. However, we are optimistic that the democratisation of 3D model generation will be a value-generating technology for the large majority. The illumination prior might be biased since most of the available and used HDR images were captured in Europe.

**Acknowledgments and Disclosure of Funding**   James Gardner was supported by the EPSRC Centre for Doctoral Training in Intelligent Games & Games Intelligence (IGGI) (EP/S022325/1). We would like to thank the attendees of Dagstuhl Seminar, 22121 - 3D Morphable Models and Beyond, for their valuable insights and discussions around this work.

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
