# Rotation-Equivariant Conditional Spherical Neural Fields for Learning a Natural Illumination Prior -Supplementary Material-

**James A. D. Gardner**
Department of Computer Science
University of York
York, United Kingdom
james.gardner@york.ac.uk

**Bernhard Egger**
Cognitive Computer Vision Lab
Friedrich-Alexander-Universität
Erlangen, Germany
bernhard.egger@fau.de

**William A. P. Smith**
Department of Computer Science
University of York
York, United Kingdom
william.smith@york.ac.uk

Here we provide further results, additional training information, and explore the how RENI performs when fit to illumination maps that contain unnatural illumination conditions.

## A  Additional Training Details

### A.1  Cosine Loss

When optimising just the latent codes to unseen environment maps, we found improved performance when including a cosine similarity loss. As with $\mathcal{L}_{\text{recon}}$ this was weighted by the sine of the polar angle, $\theta(\mathbf{d})$, to compensate for the irregular sampling of equirectangular images. Each training sample comprises $P$ pairs of directions and corresponding normalised log(HDR) RGB colours that we store in the matrices $\mathbf{D}_i = [\mathbf{d}_{i1}, \ldots, \mathbf{d}_{iP}] \in \mathbb{R}^{3 \times P}$ and $\mathbf{C}_i = [\mathbf{c}_{i1}, \ldots, \mathbf{c}_{iP}] \in \mathbb{R}^{3 \times P}$ respectively. $\mathcal{L}_{\text{Cosine}}$ is therefore defined as:

$$\mathcal{L}_{\text{Cosine}} = \sum_{i=1}^{K} \left( 1 - \frac{1}{P} \sum_{j=1}^{P} \sin(\theta(\mathbf{d}_{ij})) \frac{f(\mathbf{d}_{ij}) \cdot \mathbf{c}_{ij}}{\max \left( \|f(\mathbf{d}_{ij})\|_2 \cdot \|\mathbf{c}_{ij}\|_2, \varepsilon \right)} \right) \quad (1)$$

where $K$ is the number of training samples.

We used a value of $\varepsilon = 10^{-20}$ to prevent singularities. This loss is used again during the inverse rendering task. However as the images used in that loss are not equirectangular the $\sin(\theta(\mathbf{d}))$ term is not included.

36th Conference on Neural Information Processing Systems (NeurIPS 2022).

### A.2 Gamma Correction

For display, all linear HDR images $\mathbf{I}$ had their gamma adjusted using the following process:

1. Adjust exposure to set the white level to the $p$-th percentile ($p = 98$)

$$\mathbf{I} \leftarrow \frac{\mathbf{I}}{\text{percentile}(\mathbf{I}, p)}$$

2. Clamp between $[0, 1]$

$$\mathbf{I} \leftarrow \text{clamp}(\mathbf{I}, 0, 1)$$

3. Apply gamma correction using the standard sRGB gamma curve:

$$\gamma RGB(\mathbf{I}) = \left\{ \begin{matrix} 12.92\mathbf{I} & \mathbf{I} \leq 0.0031308 \\ 1.055\mathbf{I}^{1/2.4} - 0.055 & \mathbf{I} > 0.0031308 \end{matrix} \right\}$$

such that:

$$\mathbf{I} \leftarrow \gamma RGB(\mathbf{I})$$

## B  Additional Results

We include additional qualitative results of the RENI model. Figures 1 and 2 show further examples from the selection of test images demonstrating how RENI performs against SH and SG for increasing latent dimensionality. Figure 3 demonstrates performance differences between RENI and SH at the environment completion task. RENI's prior on natural illumination enables significantly more realistic results and no over-fitting to the provided cutout. Figures 4 and 5 show a $D = 108$ model fitting to unseen images that contain unnatural illuminations, e.g. artificial illuminations or indoor environments. Figure 6 shows a comparison between RENI and SG in the inverse rendering task and Figure 7 shows one of RENI's failure cases.

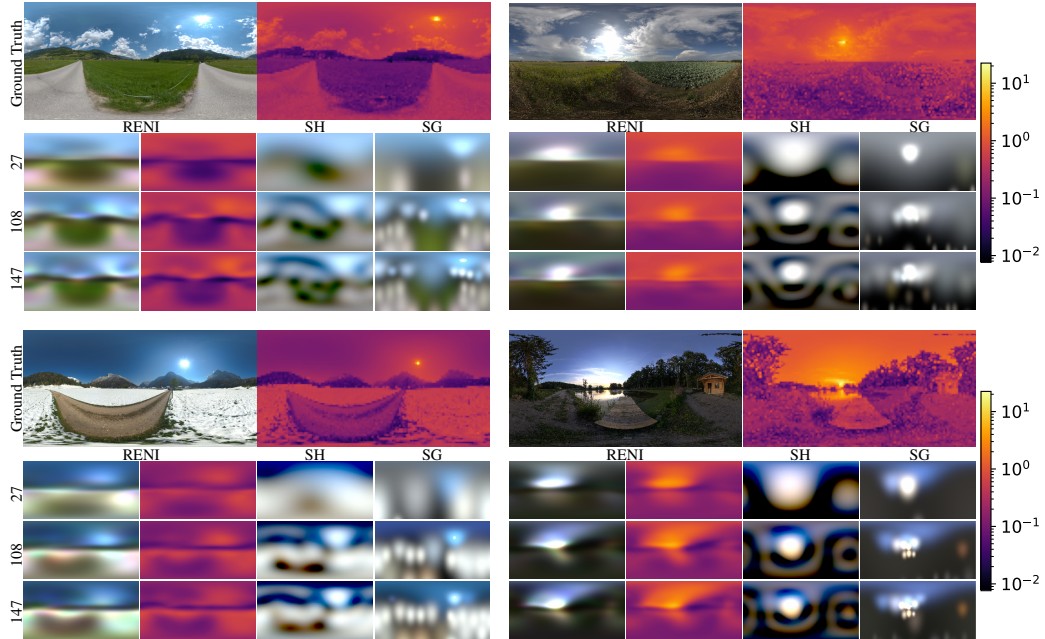

Figure 1: Generalisation to unseen images with latent code dimensions, $D = 3N$ for $N = 9, 36, 49$ and for SH of equal dimensionality (orders 2, 5, and 6). SG results are with dimensionality $D = 30, 108, 150$. Heat maps with log-scale colour bars for ground truth and RENI are also shown.

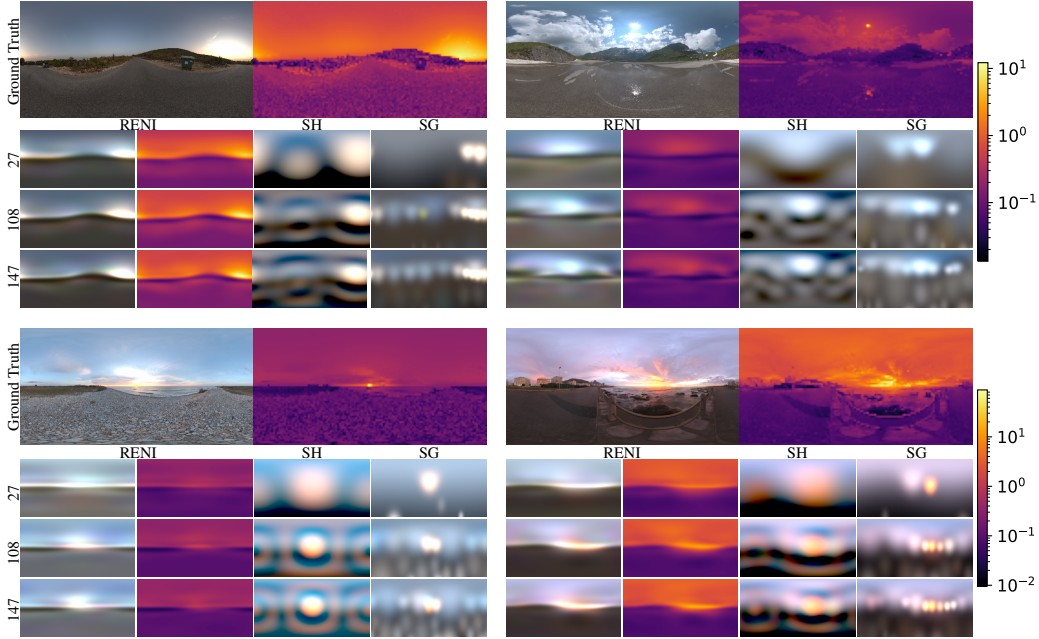

Figure 2: Generalisation to unseen images with latent code dimensions, $D = 3N$ for $N = 9, 36, 49$ and for SH of equal dimensionality (orders 2, 5, and 6). SG results are with dimensionality $D = 30, 108, 150$. Heat maps with log-scale colour bars for ground truth and RENI are also shown.

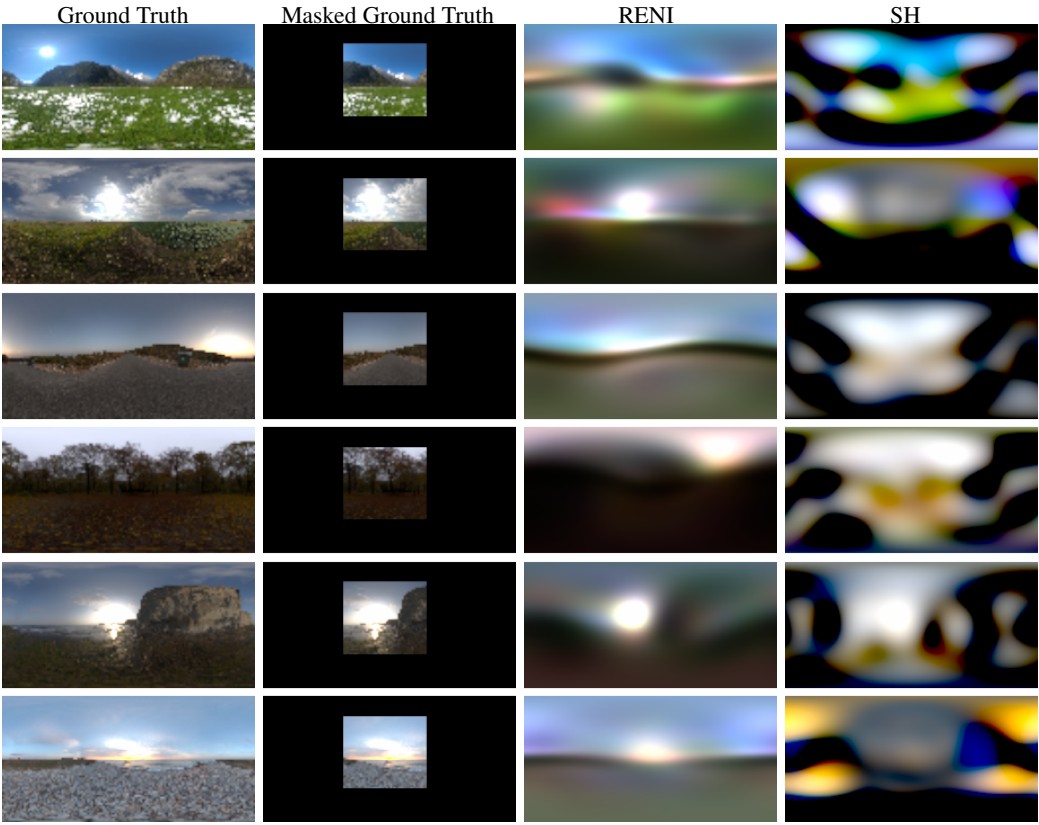

Figure 3: Fitting RENI and SH of latent dimension $D = 108$ to masked ground truth images. RENI realistically completes the environment whilst SH produces noisy, unrealistic results and overfits to the training data.

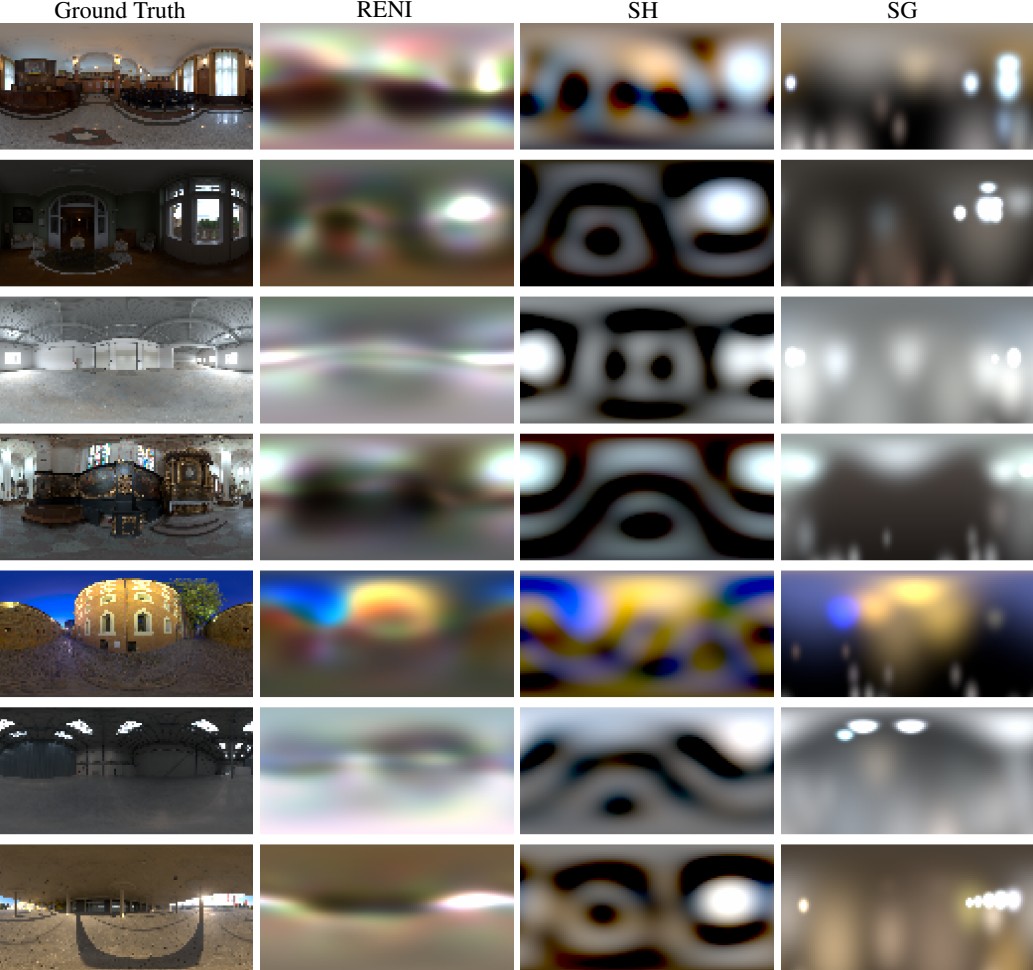

Figure 4: Fitting to images containing unnatural illumination conditions. Whilst these images are out-of-distribution for RENI, its latent space is expressive enough to still reasonably approximate some of these environments. RENI, SH and SG fit using models with $D = 108$.

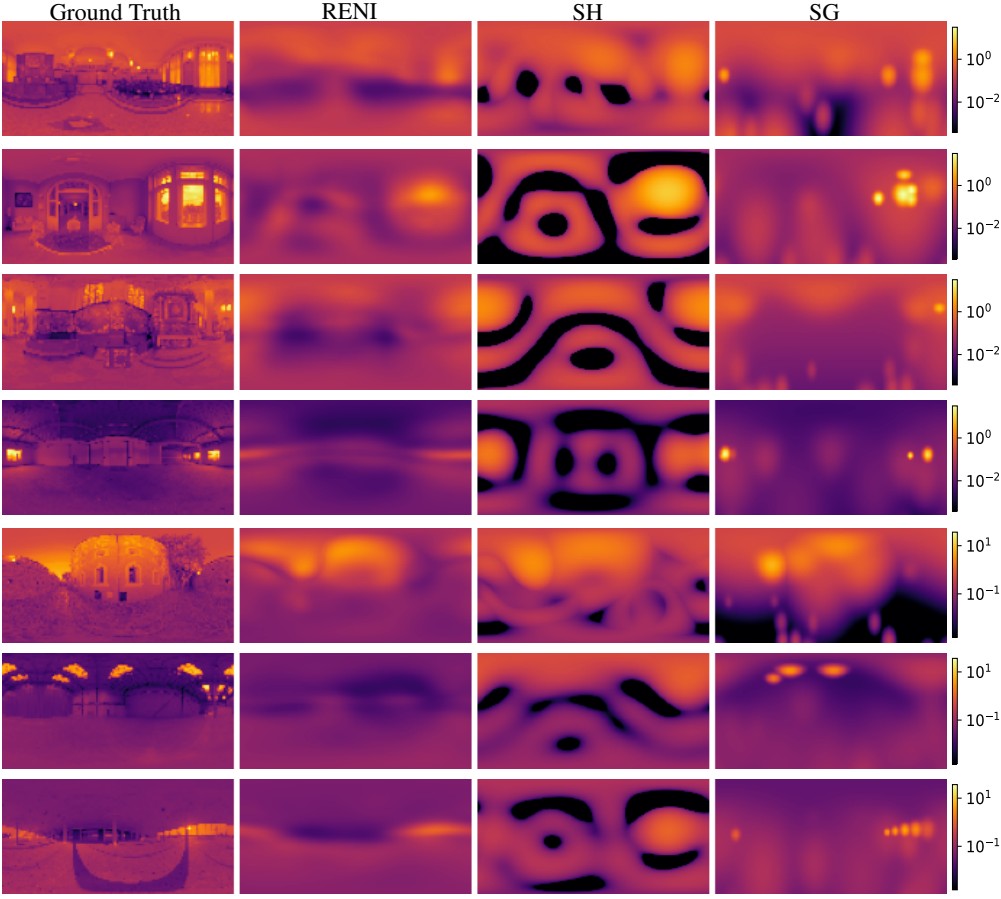

Figure 5: Heat maps with log-scale colour bars for images containing unnatural illuminations. RENI, SH and SG fit using models with $D = 108$.

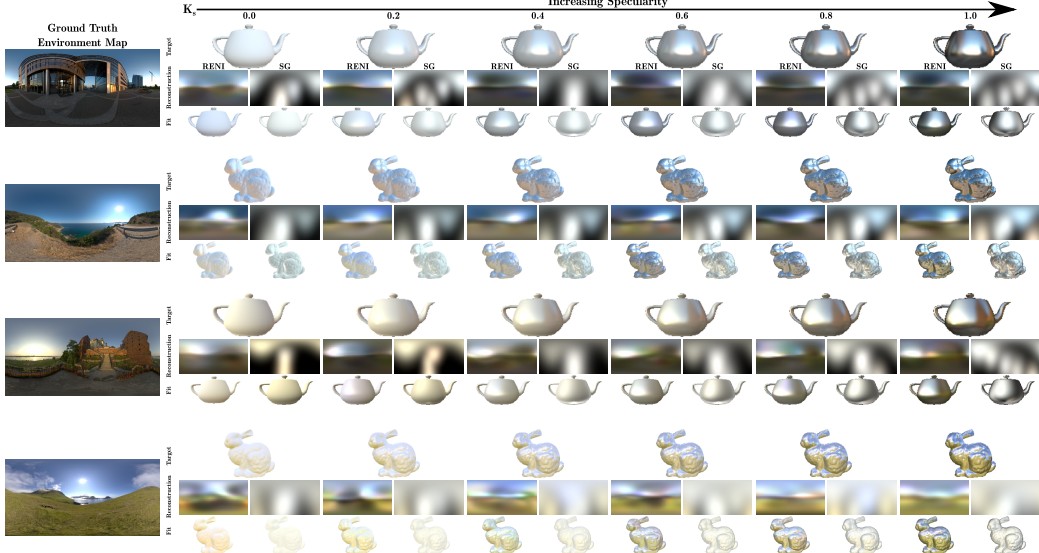

Figure 6: Reconstruction results in an inverse rendering task. The specular Blinn-Phong term $K_s$ increases from left to right in steps of $0.2$. RENI has a dimensionality of $D = 27$ and SG a dimensionality of $D = 30$. RENI outperforms SG whilst also producing environment maps that are plausible natural illumination environments as opposed to SG which produces arbitrary illuminations.

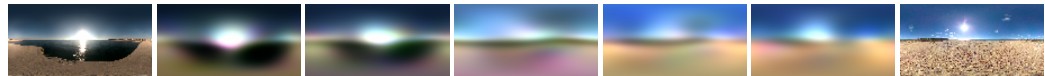

Figure 7: A failure case when interpolating between two training images using the $D = 108$ model. A reddish colour appears on the ground in the centre image that is not present in either the source or target images. This may be due to our use of HDR images. The source and particularly target have very bright sun regions; even though our loss is in log space, this likely dominates the reconstruction meaning slight colour tone errors in the darker regions are not penalised. Including the cosine loss function during training might help resolve this.