# OpenReview forum: "Rotation-Equivariant Conditional Spherical Neural Fields for Learning a Natural Illumination Prior"
_NeurIPS.cc/2022/Conference — NeurIPS 2022 Accept_

### Official Review · Reviewer_PVj2 · 2022-07-09

**Rating:** 6
**Confidence:** 4
**Soundness:** 3 good
**Presentation:** 3 good
**Contribution:** 2 fair

**Summary:**

This paper proposes a generative model for natural illumination using a neural field representation based on a variational auto-decoder. The key idea is that natural illumination is highly structured (e.g., lighting comes from above), which has a prior that can be learned and represented well by a generative model. Also, natural illumination generally has a geometric symmetry that a rotation with respect to the vertical axis (i.e., gravity axis) is equally likely, which can be used to restrict the possible illumination space. The proposed method is based on this property and is rotation-equivariant by adding a rotation-invariant transformation on the input direction and the feature vector before the network. The method is demonstrated by extensive evaluations both qualitatively and quantitatively.

**Questions:**

- Figure 1 shows the result with 3 x 20 latent code, but all the results used in Section 4 (Evaluation) has different latent dimensions (i.e., N = 9, 36, 49, 100). Is there a special reason why Figure 1 has a different dimension? Also, Figure 3 does not show the case of N = 100, which is in the quantitative comparison in Table 1.
- What is the resolution of the environment maps that are used in the training? Do all the images have the same resolution? If not, how is it sampled in the training?

**Limitations:**

The limitations and impact are well described including the memory footprint of the gram matrix, and the possible misalignment of the y-axis in the image. Broader impact and possible bias in the illumination data (i.e., captured in Europe) are also addressed properly.

**Strengths And Weaknesses:**

### Strengths
- This paper is well written. It is a good primer for understanding the structure of natural illuminations and their representation including Spherical Harmonic (SH), Spherical Gaussian (SG). The motivation of rotation invariance and equivariance is well presented, and the core idea of SO(2) equivariance is described intuitively and clearly.
- Promising results. The proposed method shows promising results both qualitatively and quantitatively as shown in Table 1 and Figures 3-5. Figure 3 shows that the proposed method RENI captures high-frequency detail such as the primary light source (i.e., sun) more accurately compared to SH and SG where the sun is more blurry. The interpolation and inpainting results in Figures 3 and 4 are also promising and interesting.
- Extensive evaluation. The paper provides extensive evaluations including interpolation, inpainting, and inverse rendering, as well as the comparison to other methods (e.g., SH, SG). The experiments are also done with different dimensions and comparisons are conducted accordingly with various latent dimensions.

### Weaknesses
- Ablation study is missing. While the paper provides extensive evaluations of a range of tasks, an ablation study is missing. A comparison to the baseline model without rotation-equivariance would help show the efficiency of the proposed method in restricting the space of natural illuminations.
- Some preliminaries can be explained more in detail. As the proposed method has some overlap with Vector Neurons (e.g., invariant layer), adding preliminaries regarding Vector Neurons would help readers better understand the proposed method.
- Reference format. Although it is not directly related to the proposed method, the reference section should be fixed. Most references miss the name of the conference or journal, and the names of authors are replaced by “et al”, which should be fixed to the full list of the authors.

---

> ### Author Response · Authors · 2022-08-02
> **Response to Reviewer PVj2**
>
> We sincerely thank the reviewer for the constructive feedback. We hope our responses address the reviewer's questions and concerns.
>
> ***
>
> ## 1. Ablation without rotation equivariance.
>
> Here we provide results for models with SO(2), SO(3) and no equivariance at three sizes of latent code dimensions $D$. For the model with no equivariance, we augmented the dataset with rotations of the images at increments of $0.785 rad$ for a training dataset size of 13384 images. The SO(2) case performs best for all latent code sizes, and both the SO(2) and SO(3) outperform the model trained purely using augmentation whilst using significantly less data.
>
> **Table-1** *The mean PSNR on the test set for models with varying levels of equivariance. Error calculated in LDR sRGB space.*
>
> | Equivariance |  D = 27 | D = 108 | D = 147 |
> | ------------------- | -------------- | -------------- | -------------- |
> |     None     |     11.32      |     15.85      |     14.64      |
> |     SO(2)    |  **17.02**  |  **19.58**  |  **19.97**  |
> |     SO(3)    |     14.00      |     18.27      |     17.45      |
>
> ***
>
> ## 2. More preliminaries.
>
> We agree this would be a good addition and will look to add this to the supplementary. We have open-sourced the code and models and hope this will allow the easier development of derivative works and enable RENI to function as a plug-in replacement for SH and SG in many inverse rendering pipelines.
>
> ***
>
> ## 3. Mistakes in references.
>
> Thank you for highlighting this issue. This has now been fixed with all references now showing the full list of authors' names and any that were missing the name of the conference or journal have been updated.
>
> ***
>
> ## 4. Choice of latent code dimensions in Figure 1 and Figure 3.
>
> The choice of latent code dimension of $3 \times 20$ in Figure 1 was an aesthetic choice to balance a high number of vectors in the plot of the latent code without over-crowding. We did not include the results for this latent dimension size in the results Table 1, due to it not being an exact size match for SH, the nearest of which would be a 4th order SH with $N = 25$. The choice not to include the $N = 100$ case in Figure 3 was made to reduce the size of the figure.
>
> ***
>
> ## 5. Resolution of environment maps.
>
> The full resolution of all the environment maps used during training and testing is $64 \times 128$. However, during training, we use a progressive training regime incrementally increasing the resolution of the images from $16 \times 32$ up to the final full resolution. This multi-resolution training enables the network to quickly learn the low-frequency content of the images early in training and progressively fit to higher frequency content.

---

> > ### Comment · Reviewer_PVj2 · 2022-08-09
> > **Response to authors**
> >
> > Thank you for the response and the updated manuscript. The rebuttal clearly addresses my concerns, including ablation without equivariance (at different latent code dimensions as well), and implementation details about the choice of latent code dimensions and resolution of environment maps that can help reproduce the paper. The reference format is improved, but it still needs to be fixed (e.g., “et al” in [18], unnecessary {Neurips}, _eprint, and so on). In any way, thank you for the thorough response, my major concerns are clearly addressed, so I would keep my rating.

---

### Official Review · Reviewer_VopV · 2022-07-11

**Rating:** 6
**Confidence:** 2
**Soundness:** 3 good
**Presentation:** 3 good
**Contribution:** 3 good

**Summary:**

The paper proposes a generative model for illumination (spherical incident radiance fields of incoming light from sources at infinite distance, not interacting with the scene). It uses a vector-neuron-based network (encoding the illumination function in MLPs with SIREN-activations), which provides SO(3)-equivariance (of which the SO(2)-part is used/relevant in practice) which is trained as a variational "auto-decoder", a variant of a VAE that does not require the encoding part (which is harder to implement in an equivariant way). The model is trained on a set of HDR environment maps and can later be used for inference task. Applications demonstrated include inverse rendering of objects with local illumination models.

The main contribution seems to be the design of a network representing a statistical illumination prior that is also equivariant, i.e., can learn lighting scenarios under various rotations without (cumbersome) data augmentation. According to the paper (I am not familiar enough with the literature to confirm this), only few pieces of prior work have considered priors on illumination conditions but had less sophisticated solutions in terms of representation and handling large dynamic range.

**Questions:**

If used in an inverse-rendering scenario (say, reconstruction from multi-view photography), what would be required (in addition) to employ the method)? As far as I could see, occlusion and interreflection effects are not modeled (which is fine, but it might be interesting to understand if the method is directly applicable to more challenging tasks in practice).

**Limitations:**

I have not found major limitations that are not discussed adequately. The discussion of broader social impact also appears adequate to me.

**Strengths And Weaknesses:**

**Strength:**
- The paper proposes a sophisticated system with many complex parts that are carefully tuned to fit together. In particular, maintaining equivariance at all stages involves some non-straightforward and recent techniques.
- Handling SO(3)-invariance is hard (and group-theoretic "brute-force" solutions based on Wigner-Matrices easily become conceptually very challenging), and the proposed approach appears to navigate around the difficulties well, providing a practical solution, while still remaining expressive and accurate.
- The method apparently does not have fixed resolution limits that linear representations would be prone to (even if just serving as output layer of a more complex network). Functional complexity is still limited by scaling issues (Gram matrix).
- Results are good, surpassing linear SH and non-linear SG base-lines numerically. Visual results are also plausible/convincing.
- The paper is very well written.

**Weaknesses:**
- The paper is mostly a "systems" approach to solve a specific application problem. While very well executed, the broader impact (methodological novelty) to machine learning is limited (although non-negligible, as generative models on spherical domains might be a relevant class of problems). One could debate whether a graphics or vision venue would be a better fit for this topic.
- Quantitative evaluations are limited to SH/SG base-line methods; however, it is probably difficult to find better comparisons due to limited prior work and complexity of the overall setting.
- Despite good writing, the compositional nature of the contribution requires studying the background literature (in particular, vector neurons and auto-decoders) in depth to get a full picture. Again, this is probably hard to avoid.
- The contribution is to some extend incremental (but solid, nonetheless).

---

> ### Author Response · Authors · 2022-08-02
> **Response to Reviewer VopV**
>
> We sincerely thank the reviewer for the constructive feedback. We hope our responses address the reviewer's questions and concerns.
>
> ***
>
> ### 1. More suited to a graphics or vision venue.
>
> While we agree the work could potentially appeal to a  graphics or vision audience, we feel the impact of a rotation-equivaraint generative model for spherical signals may lay beyond just graphical applications. In addition, we believe there is machine learning methodological novelty in our framework. Finally, much of the recent work on distributed representations for inverse rendering has been published at NeurIPS.
>
>
> ***
>
> ### 2. Requires studying the background literature.
>
> We agree with the reviewer and hope that by our sharing of the source code and models it will enables easier development of derivative works and enable RENI to function as a plug in replacment for SH and SG in many inverse rendering pipelines.
>
> ***
>
> ### 3. The contribution is to some extend incremental.
>
> We agree with the characterisation that a generative model for spherical domains is an interesting problem class but believe that this does amount to methodological novelty in machine learning. No one has used the vector neurons framework for representing domains other than point clouds to the best of our knowledge. We are, therefore, the first to do so for image data (specifically spherical images using a directional neural field) and propose the more complex variant to handle SO(2) equivariance.
>
> ***
>
> ### 4. Additions required to use in a more complex inverse rendering scenario.
>
> Occlusions and inter-reflection effects are not handled by an illumination model, but by the way it is applied in the scene i.e. rendering of global vs local illumination. Both are possible using RENI and for complex rendering we would also need explicit modeling of material etc. Whilst this is not the case in many frameworks we are convinced that our model will be a key component to go in this direction as the particular weakness of SH and SG is specularities.

---

### Official Review · Reviewer_cfSy · 2022-07-11

**Rating:** 3
**Confidence:** 4
**Soundness:** 2 fair
**Presentation:** 3 good
**Contribution:** 2 fair

**Summary:**

This paper presents the neural illumination model based on the variational auto-decoder. By introducing the rotational invariance to the latent variable, the model can represent plausible environment map that could be useful for the inverse rendering task. The evaluation showed that the proposed model has better representation ability with a compact latent code.

**Questions:**

- Please explain more clearly what new ideas in the paper could be useful in the community.
- Please clarify more how to represent the high-frequency lighting information with the proposed method. Also, please provide more theoretical evidence of how useful individual components in the proposed method (i.e. latent representation and rotational invariance) are in inverse rendering in practice.
- If there is a misunderstanding, please correct me on the concerns pointed in weakness.


**Limitations:**

The authors describe limitations and negative societal impact.

**Strengths And Weaknesses:**

Strengths:

- The attempt to represent light sources in a generative model is very interesting. Adding rotational invariance to the latent space in a generative model is also an interesting attempt.
- The authors properly addressed potential limitations in the method.
- The description of the proposed method is clear and almost no ambiguity exists.

Weakness:

This research consists of two elements: defining the lighting with a generative model and making the model rotationally invariant (around the gravity axis). Therefore, I would like to address concerns about each of these factors.

- As for the first part, instead of directly optimizing the pointwise function, learning a latent space (i.e. generative model) for a target domain from external data in advance and optimizing its latent variables to generate an instance that satisfies specific criteria have already been done in many computer vision tasks such as CNN-SLAM and StyleNeRF. Therefore, essentially using a similar strategy to generate an environment map doesn’t seem to be a very novel idea. Although an embedded lighting space can indeed represent a wide range of environment maps with fewer parameters, the spaces generated in this way are generally limited to the low-to-mid-frequency range as was mentioned in the paper. Nevertheless, there is little effort to express high-frequency components, as has been done in recent generative models such as StyleGAN. Furthermore, I feel that claiming a contribution for the learnt latent space as “natural illumination prior” is somewhat of an overclaim, as it is true of all data-driven models including other than generative models.

- As for the rotational invariance, I recognize the advantage of being able to learn a latent space with a small number of data, including overlap due to rotation, but I feel that the negative effects of this constraint are not small. As authors have properly identified this in the paper, in real inverse rendering applications, it is very unusual that the camera is parallel to the ground, and if we want to use the environment map generated by the proposed method, we need to compensate for the gravity direction for the coordinate system transformation. But not all digital cameras unfortunately have a gyro sensor. In addition, the realization of rotational invariance is almost entirely based on existing methods, except for the transfer of coordinates to angle representation, so there seems to be little novelty in itself though it seems novel that this invariance is plugged in the generative mocel.

- Due to the lack of real experimental results about the inverse rendering task, it is not clear how useful this model in the real applications (e.g., how the reconstruction accuracies of other attributes such as shape and reflectance improve). In addition to synthetic examples as presented in the paper, there should have been results showing the superiority of the proposed method in real experiments.

- Authors claimed that the O(n) trick proposed in [15] can resolve the scalability issue but have not been verified in the paper. The paper also didn’t show the comparison between SO(2) and SO(3) invariance.

- There are too many mistakes in references.

---

> ### Author Response · Authors · 2022-08-02
> **Response to Reviewer cfSy 1/2**
>
> We sincerely thank the reviewer for the constructive feedback. We tried our best to address the reviewer's concerns and questions and hope the reviewer finds our responses satisfactory.
>
> ***
>
> ### 1. The novelty of idea in relation to StyleNeRF and CNN-SLAM.
>
> We could not see the relation of CNN-SLAM to generative models. If the reviewer can add an additional explanation, we are happy to discuss it further. However, taking the general point that building a generative model of illumination environments is no different to building generative models of other domains such as images or NeRFs, we slightly disagree. Illumination environments are spherical and have a high dynamic range. These two properties require special handling. A Spherical CNN [1] based GAN with an HDR loss would, to the best of our knowledge, be a novel solution in itself. However, we took a neural fields based approach because it enables rotation equivariance in the representation. It also avoids having to choose the sample grid in advance, and therefore 1. has the potential to more compactly represent the spherical signal by assigning network capacity adaptively and 2. allows multiresolution training with a single network.
>
> ***
>
> ### 2. Lack of work done to address the loss of high-frequency components.
>
> We did implement and test a variant of RENI that used FiLM-Conditioning in the hopes that this model would be capable of capturing higher frequency signals. However, we found this not to be the case due, we believe, to the small size of our dataset. An alternative we are interested in trying in future work will be to implement this rotation-equivariant spherical neural field in a GAN framework. Using a spherical CNN as the discriminator will help the model capture higher frequency, more detailed images.
>
> ***
>
> ### 3. Our claim of a natural illumination prior.
>
> We have an explicit prior over natural illumination since we assume the elements of $\mathbf{Z}$ are normally distributed (line 143), and our decoder learns the nonlinear mapping from this latent space to the spherical signal value. We can therefore sample from this prior distribution to generate realistic illumination environments or use a prior loss on an estimated latent code to regularise inverse tasks.
>
> Whilst other generative models, e.g. StyleGAN, when trained on images of the natural environment, would also inherently capture statistical regularities of the illuminations present in those images, these would require large amounts of data, something prohibitively challenging to obtain for HDR equirectangular environment maps, and would not be rotation-equivariant.
>
> By training on HDR equirectangular images and designing our model to be rotationally equivariant, our model can easily replace SH and SG as the representation of distant illumination in many inverse rendering tasks. Furthermore, when sampled, our model will only produce plausible environment maps, which is highly useful for resolving albedo-illumination ambiguities in inverse rendering. We feel this combination of benefits warrants using the term 'natural illumination prior'.
>
> ***
>
> ### 4. Negative consequences of rotational invariance.
>
> The reviewer raises a valid point that not all digital cameras have a gyro sensor; however, we believe that in such a case, this could be compensated for via external estimation of the gravity vector using a method similar to [2] or via explicit optimisation of the rotation angle during inverse rendering. Our key motivation though is that the captured environments themselves do have a canonical up direction and that by respecting this we learn a more parsimonious and expressive model.
>
> ***
>
> ### 5. Lack of real-world experiments.
>
> The core proposition of our paper was to introduce a new generative model for spherical signals and that a demonstration of its use in a simple inverse rendering problem would suffice to show its value in this domain. We assume that RENI will perform comparably or superiorly in any framework where SH or SG is used (assuming that the illumination is natural). We spent quite some time deciding how to tell the story in this paper and thought it would be most effective if isolated from other potential extensions and applications. This aids in keeping the story clean and not diluted in further complexity and choices of frameworks that RENI could be combined with. We do plan in future work to implement RENI in more substantial inverse rendering applications with real-world 'in-the-wild' images.

---

> > ### Author Response · Authors · 2022-08-02
> > **Response to Reviewer cfSy 2/2**
> >
> > ### 6. Ablation of equivariance.
> >
> > Here we provide results for models with SO(2), SO(3) and no equivariance at three sizes of latent code dimensions $D$. For the model with no equivariance, we augmented the dataset with rotations of the images at increments of $0.785 rad$ for a training dataset size of 13384 images. The SO(2) case performs best for all latent code sizes, and both the SO(2) and SO(3) outperform the model trained purely using augmentation whilst using significantly less data.
> >
> > **Table-1** *The mean PSNR on the test set for models with varying levels of equivariance. Error calculated in LDR sRGB space.*
> >
> > | Equivariance |  D = 27 | D = 108 | D = 147 |
> > | ------------------- | -------------- | -------------- | -------------- |
> > |     None     |     11.32      |     15.85      |     14.64      |
> > |     SO(2)    |  **17.02**  |  **19.58**  |  **19.97**  |
> > |     SO(3)    |     14.00      |     18.27      |     17.45      |
> >
> > ***
> >
> > ### 7. Mistakes in references.
> >
> > Thank you for highlighting this issue. This has now been fixed with all references now showing the full list of authors' names and any that were missing the name of the conference or journal have been updated.
> >
> > ***
> >
> > ### 8. Please explain more clearly what new ideas in the paper could be useful in the community.
> >
> > This work is the first example of a rotation-equivariant generative model for spherical signals. No one has used the vector neurons framework for representing domains other than point clouds to the best of our knowledge. We are, therefore, the first to do so for image data (specifically spherical images using a directional neural field) and propose the more complex variant to handle SO(2) equivariance. The other reviewers seem to agree with our view that this model is useful for the community.
> >
> > For inverse rendering, our primary requirement is a low dimensional parametric representation (since it is these parameters we will have to optimise at test time) that leads to renderings with low error. We do not necessarily need a perfect reconstruction of the environment image. Our generative model for illumination environments can also be used to quickly generate realistic synthetic data, which can be used to make other models more robust to varying illumination conditions. More generally, the same equivariant spherical generative model could find application in modelling any other spherical signal such as 360 video, geospatial data, distributions of directional data and so on.
> >
> > ***
> >
> > ### 9. Please clarify more how to represent the high-frequency lighting information with the proposed method. Also, please provide more theoretical evidence of how useful individual components in the proposed method (i.e. latent representation and rotational invariance) are in inverse rendering in practice.
> >
> > There are several approaches we see to enable RENI to capture higher frequency lighting information. For example, with a larger dataset, we expect that using a FiLM-Conditioned SIREN could enable a more expressive model. We also envisage using our rotation-equivariant conditional spherical neural fields as the generator in a GAN framework and expect this to enable the reproduction of high-frequency illumination components.
> >
> > There are many applications in inverse rendering where our latent representation and rotational invariance can be used. For example, our latent representation of natural illuminations could help resolve albedo-illumination ambiguities. When estimating a person's skin albedo from a single image, there is ambiguity between the contributions of skin colour and illumination. This ambiguity is largely unaddressed in current research, with practically all current models biasing strongly towards light skin colours. This bias results in dark skin tones being estimated as lighter skin under dark illumination. Our prior for natural illuminations could potentially alleviate this issue, either at inference time, where our prior would restrict the search space of illuminations by providing an expressive illumination model encoding only natural illuminations in a small number of latent parameters thus enabling the models to converge quickly on the most likely explanations. Or, alternatively, at training time, where our model could be used to produce large amounts of synthetic training data, where the rotation-equivariance of our model would enable simple rotations of those illumination conditions. Futhermore, as shown in Table 1, our equivariant models also produce more expressive latent spaces with significantly less data and time required to train.
> >
> > ***
> >
> > [1] Taco S. Cohen, Mario Geiger, Jonas K ̈ohler, and Max Welling. Spherical CNNs. In International Conference on Learning Representations, 2018
> >
> > [2] Wenqi Xian, Zhengqi Li, Matthew Fisher, Jonathan Eisenmann, Eli Shechtman, and Noah Snavely. Uprightnet: Geometry-aware camera orientation estimation from single images. In Proceedings of the IEEE/CVF International Conference on Computer Vision, 2019.

---

> > > ### Comment · Reviewer_cfSy · 2022-08-08
> > > **Response to Authors**
> > >
> > > Thank you very much for your very thorough response to my concerns. I read all the responses from authors and other reviewers' comments carefully and become more positive about this work. Especially, the comparison of SO(2) over SO(3) equivariance was very helpful because more clarity on the value of providing rotational equivalence to the generative model of the illumination map. I agree that the presence or absence of gravity direction is not a fundamental issue either.
> > >
> > > I have to apologize that I incorrectly mentioned CNN-SLAM but what I should have mentioned was Code-SLAM (Bloesch2018). This method does not pixelwisely regress depth values from the image, but rather generates the only plausible depth map from compact latent variables by optimizing them with a pre-trained decoder, which I believe has some similarities in terms of the methodology and motivation with the proposed method. Also, the fact that the proposed method relies strongly on vector neurons, though the domain is not a point cloud, is still a concern.
> > >
> > > But more than novelties, my major concern still lies in the lack of real evaluation. Though the information given by the authors about other inverse rendering tasks was promisingly useful, I still wish they had been properly verified in the submitted paper. It is often said that what works in theory, can often not work in reality. Of course, it is not wrong to emphasize the clarity of the storyline, but I believe evaluation on the real data is also important, especially for the learning-based approach which easily overfits the specific domain of data. The discussion about the high-frequency components and the benefit of natural illumination prior is somewhat persuasive, but I also think it should have been actually verified.
> > >
> > > In any case, the authors have given me great clarity on many of my concerns. Thank you very much for your very thoughtful response. I will carefully discuss in the post-rebuttal phase whether or not to change my evaluation.

---

> > > > ### Author Response · Authors · 2022-08-09
> > > > **Response to Reviewer cfSy**
> > > >
> > > > We are pleased that our previous replies helped to clarify and promote our contribution and hope our further responses help address the reviewer's remaining concerns.
> > > >
> > > > ***
> > > >
> > > > ### Similarity to Code-SLAM
> > > >
> > > > Code-SLAM learns a generative model (decoder) of depth maps for indoor scenes. It does so using a convolutional architecture. Hence, a given latent code defines a complete depth map of fixed resolution via the series of convolutional layers. There is no invariance or equivariance built into the representation. On the other hand, our generator is coordinate-based (more precisely, direction-based). For any direction and conditioning latent code, it outputs the value of the signal in that direction. Using a coordinate-based representation means we do not have to choose or fix a specific image resolution in advance (enabling multiresolution training using the same single network). Also, it is this construction that enables the rotation-equivariant representation. So, we see the relevance of Code-SLAM but also argue that our contribution is significantly different.
> > > >
> > > > ***
> > > >
> > > > ### Reliance on vector neurons
> > > >
> > > > We want to take this opportunity to further emphasise our difference with vector neurons. Their approach was designed and envisaged for point cloud data. For classification or segmentation architectures, their input is a variable length list of 3D points, and they build upon either PointNet or DCGNN architectures. For implicit neural reconstruction, they build an occupancy network conditioned on a vector neuron latent code such that an occupancy probability is outputted for any point in 3D space. Our set-up is quite different but inspired by the rotation equivariance of the vector neuron representation. We represent a continuous spherical (directional) signal (i.e. an image), seek a different (SO(2)) rotation equivariance, and our coordinate inputs are unit vectors instead of points in 3D space. We do not see reliance on the vector neuron representation as a concern. Instead, we see it as a novel extension of a clever idea that has not been widely picked up by the community yet. Another way to view this is that every other neural network based method uses scalar neurons, but we do not consider this alone as a reason to cite a lack of novelty between methods.
> > > >
> > > > ***
> > > >
> > > > ### Lack of real evaluation
> > > >
> > > > We have shown that our model can better represent natural illumination environments using the same parameters as the most widely used representations (spherical harmonics and spherical Gaussians). We have also shown that the latent space is well behaved when optimising for illumination, either to fit the model directly to a partial or complete environment map or when solving an inverse rendering task with other parameters known. We believe that these two conclusions already demonstrate that replacing SH or SG lighting with RENI in an inverse rendering problem will improve performance. The model's capability to better represent lighting means that the potential accuracy of the estimated lighting is higher. Due to the entangled nature of the inverse rendering problem, improving one of geometry, material properties, or lighting means that the estimation of the other two quantities will also be more accurate. While we agree that improving the reconstruction of high-frequency details remains an open problem, even without this, our lighting approximation is better than SH or SG, so we believe our conclusion already holds. We also agree that including further, more challenging inverse rendering problems would be a good addition to our evaluation, but within the 9-page limit did not feel this was possible while still doing justice to explaining our method (and note that another reviewer felt we should have included more description of supporting background methods).

---

### Official Review · Reviewer_1WWp · 2022-07-14

**Rating:** 7
**Confidence:** 5
**Soundness:** 3 good
**Presentation:** 3 good
**Contribution:** 3 good

**Summary:**

The paper proposes a neural representation based on spherical neural fields
for natural illuminations that are compact and rotation equivariant.
It proposes a method based on variational auto-encoders to learn such a
representation. The learned model can act as a statistical prior for natural
illuminations. The experiment results show that the proposed neural
representation can better represent natural illuminations than traditional
spherical lighting representations such as spherical harmonics and spherical
gaussian. The paper also shows that the learned model can be incorporated in
inverse renderings tasks as illumination priors and leads to better lighting
estimation.


**Questions:**

See above.

**Limitations:**

The limitations are well discussed.

**Strengths And Weaknesses:**

Strengths

1. The paper proposes a novel solution to an important problem. The idea of
using spherical neural fields to represent natural illuminations is inspiring,
and the design of the input to make the represent rotation equivariant is
also technically sound. The paper also proposes to apply variational
auto-encoders to learn such a model, which also makes sense to me.

2. The paper performs a thorough evaluation against baseline methods to show the
advantages of the proposed representation. It also performs different ablation
studies to validate the capability of the representation and different design
choices.

3. Such a representation has great potential in many applications, especially in
inverse rendering tasks. The paper does experiments which show that the learned
priors help estimate more accurate lighting.

Weakness:

1. The paper evaluates the performance of the model with different latent code
size. I am also wondering how the performance changes with different sizes of
the spherical neural field network. This is an important factor because ideally,
we would like the network to be as small as possible so that the cost of
evaluating such a network would be minimal, especially when using it in inverse
rendering tasks.

2. In Table 1, the performance of SH drops when D increase from 147 to 300. What is
the reason for it?

3. In the third row of Figure 4, why there will be reddish colors that do not
exist in the two source environment maps?

4. In the inverse rendering tasks, the paper only compares to spherical
harmonics. It's also worth adding comparisons to spherical gaussians as they are also
commonly used in inverse rendering tasks as in [60].

5. When fitting the environment maps in Figure 3, it seems that the performance
of the proposed method does not improve much with the increase of the coding
size, while SG and SH can better approximate the shape of the lighting. What
are the limiting factors that prevent the method from further reconstructing more
accurate environment maps? How such a problem can be tackled?


Overall, I like the idea of the paper, and the evaluations are also thorough. I
believe that such an illumination prior can be useful in many tasks such as
inverse rendering and is worth introducing to the community.

---

> ### Author Response · Authors · 2022-08-02
> **Response to Reviewer 1WWp**
>
> We sincerely thank the reviewer for their constructive feedback and are pleased they see the potential of our work. We hope this response answers the reviewer's remaining questions and concerns.
>
> ***
>
> ### 1. An ablation of neural field sizes.
>
> We have run an ablation of different model sizes with an increase and decrease in the number of layers in the network. With the smaller network, reconstruction quality suffers due to the representational power of the network being reduced. Whereas the larger networks over-fit on the training data and optimising latent codes to fit unseen images becomes more challenging, perhaps due to the small size of the dataset.
>
> **Table-1** *The mean PSNR on test set for varying network and latent sizes. Error calculated in LDR sRGB space.*
>
> |        # of Hidden Layers         | D = 27 | D = 108 | D = 147 |
> | --------------------------------- | ------------- | -------------- | -------------- |
> |            3 Layers               |     16.25     |     18.29      |     18.57      |
> |            5 Layers               |  **17.02** |  **19.58**  |  **19.97**  |
> |            7 Layers               |     16.38     |     18.13      |     18.15      |
>
> ***
>
> ### 2. Drop in performance in SH from 147 to 300.
>
> This issue was due to a bug in the generation of the SH representation. Line 146 of hdri\_dataset.py file applies the sinewighting function to the equirectangular image; however, this was applied again in the function getCoefficientsFromImage() called on line 149 of hdri\_dataset.py. All results affected by this bug have been corrected in the updated version of the paper. No conclusions are altered by these changes.
>
> ***
>
> ### 3. Reddish colours in Figure 4.
>
> The reviewer is correct and we agree this is unusual. We think this may be due to our use of HDR images. The source and particularly target have very bright sun regions; even though our loss is in log space, this likely dominates the reconstruction meaning slight colour tone errors in the darker regions are not penalised. Including the cosine loss function during training might help resolve this. We have moved this example to supplementary as a failure case and replaced it in the main paper with a different example that does not have an artefact.
>
> ***
>
> ### 4. Include SG in inverse rendering comparison.
>
> We agree that this would be an good addition to the paper and have included this as Figure 6 in the updated supplementary.
>
> ***
>
> ### 5. Small performance gains for size of latent code and limiting factors in higher frequency details and approaches to solving these.
>
> First, we believe that this observation demonstrates a strength of our model. It already saturates generalisation ability with a relatively low dimensional latent space due to its nonlinear and rotation equivariant representation. In other words, our model is efficient in learning common low frequency shared components but, due to the limited size of the training set, it cannot learn any higher frequency components even with a higher dimensional latent space.
>
> However, we agree that some consideration of how to improve on this front is important and we have already begun to explore two ideas. First, as mentioned in the paper, we tried a FiLM-Conditioned SIREN and believe this should lead to better reconstruction of high frequency details (though likely still requiring a larger training dataset). Second, we believe that the introduction of a discriminator applied to the images generated by our model would help it to learn high frequency features in order to increase realism. This would require design of a rotation invariant discriminator, probably in the form of a spherical CNN [1].
>
> ***
>
> [1]  Taco S. Cohen, Mario Geiger, Jonas K ̈ohler, and Max Welling. Spherical CNNs. In International Conference on Learning Representations, 2018

---

### Meta-Review · Area_Chair_cV3Z · 2022-08-23

**Recommendation:** Accept
**Confidence:** Less certain

**Metareview:**

The paper introduces a rotation-equivariant conditional spherical neural fields for illumination priors.
Reviewers mostly like the novelty of the proposed approach, its fit for the considered task of illumination priors, technical soundness and experimental evaluation that is thorough and shows merits of the approach. The rebuttals to the reviewers also were thorough and addressed reviewers concerns well. All in all this is a conceptually and experimentally solid and interesting paper that merits publication at NeurIPS.

**Award:**

No

---

### Decision · Program_Chairs · 2022-09-14

Accept